# Efficacy of a Horticultural Therapy Program Designed for Emotional Stability and Career Exploration among Adolescents in Juvenile Detention Centers

**DOI:** 10.3390/ijerph19148812

**Published:** 2022-07-20

**Authors:** Kyoung-Hee Park, Soo-Young Kim, Sin-Ae Park

**Affiliations:** 1Department of Environmental Science, Graduate School, Konkuk University, Seoul 05029, Korea; pkh7994@daum.net; 2National Institute of Biological Resources, Environmental Research Complex, 42 Hwangyeong-ro, Seo-gu, Incheon 22689, Korea; sy7540@korea.kr; 3Department of Systems Biotechnology, Konkuk Institute of Technology, Konkuk University, Seoul 05029, Korea; 4Department of Bio and Healing Convergence, Graduate School, Konkuk University, Seoul 05029, Korea

**Keywords:** horticultural therapy, career exploration, juvenile offenders, adolescent, Korean native plants

## Abstract

We aimed to develop a horticultural therapy program to prepare adolescents at the Dae san juvenile detention center (D-JDC) for their return to society. The effects of the program on emotional stability and career exploration were investigated. Adolescents who wished to participate in the horticultural therapy program were recruited from the D-JDC. Data were collected using various questionnaires before and after the program was implemented. Thirty-five (mean age, 15.74 ± 1.65 years; 11 males, 24 females) students were enrolled. The program mainly consisted of plant cultivation activities, such as seeding, transplanting plants, cutting, harvesting, and post-harvest utilization. To evaluate emotional health, the ego-resiliency scale was used. To evaluate social behavior, the inventory of parent and peer attachment, peer attachment scale, and social skills scale were used. Career exploration was assessed using the career preparation behavior scale and the career decision-making self-efficacy-short form. Peer attachment, social skills, and career preparation behavior showed significant improvements after the program, with the students responding positively in the post-program surveys. Our horticultural therapy program helped improve the career exploration and social skills of D-JDC students and positively affected their emotional stability. Based on our findings, horticultural therapy can be used as a correctional program for adolescents in D-JDC to help them return to society.

## 1. Introduction

Adolescence is a transitional period from childhood to adulthood, during which an individual undergoes drastic physical, psychological, and social changes [1,2]. In particular, adolescents can experience emotional instability due to a lack of guardianship, poor access to education, and/or uncertain career paths, leading to social maladjustment; of note, the number of youths in detention centers is increasing [3,4]. Recently, 25% of the US news reports covered teen crimes and juvenile crimes. In China, juveniles account for 30% of all criminal offenders [5]. Although the overall number of juvenile offenders in South Korea has steadily declined over the past decade, the number of violent crimes committed by juveniles is increasing [6]. In Korea’s Juvenile Act, which is a treatment system for juvenile offenders, juveniles are specified as a person under the age of 19. Under this law, the Juvenile Departments of the Family Court and Juvenile Departments of the district courts will take special measures such as protective dispositions and criminal dispositions for environmental adjustment and character correction of antisocial juveniles. Juvenile crimes between the ages of 10 and 14 are considered juvenile protection cases, and juveniles under the age of 19 are punished as juvenile protection cases or general crime cases [6]. To prevent the increase in juvenile crimes and to decrease the prevalence of violent crimes, systematic and continuous corrective education and follow-up management are important. To this end, juvenile detention centers (JDCs) are being operated in South Korea [7,8]. JDCs are special educational institutions affiliated with the Ministry of Justice that accommodate and educate convicted adolescents aged from 10 to 19 years who are sentenced to prison time by the court’s juvenile department [2,6,9].

Depending on the educational background, age, and admission time of JDC students, they are provided specific educational programs, career development training, personality education courses, and general education. Furthermore, Dae san JDC (D-JDC) provides medical care and rehabilitation education to protected young people. Personality development education, which has been designed for the reformation of prisoners in South Korea, is mainly aimed at influencing psycho-emotional aspects such as self-esteem, return to society, mental health, and interpersonal relationships. Research trends by subject are in the order of horticultural therapy (25%), art therapy (20%), and music therapy (17.5%) [10]. In addition to existing programs, it is necessary to develop and implement specialized programs involving artistic and physical activities to enhance emotional stability and promote social reentry [5,11]. The future of juvenile offenders is not necessarily one of maladjusted adulthood or failure, and through effective probation and support, it is possible to steer juvenile offenders onto a trajectory of positive development [8,12,13].

Horticultural therapy is a professional client-centric treatment that utilizes aspects of horticulture to meet the specific therapeutic or rehabilitative needs of participants; it was first introduced in South Korea in the 1980s and has become increasingly popular since [14]. The more severe the emotional instability, the stronger the tendency to be psychologically frail and socially distant. Hence, the main goal of this therapy was to maximize social, cognitive, physical, and psychological functions and enhance general health and wellness [15]. Horticultural therapy was found to reduce aggression and excitement and increase positive thoughts in juvenile offenders, resulting in smooth interpersonal relationships and increased adaptability in school life [16,17,18,19]. This therapy was also effective in reducing stress and improving mental health [19,20,21]. A horticultural therapy program for female teenagers at JDCs was found to have a positive effect on psychological stability and personality development, including psychological maturity, self-esteem, and subjective happiness [22]. Furthermore, since the career and vocational preparations required during adolescence are limited at JDCs [23], the Teenager Occupation Program (TOP) was developed by the Korea Youth Counseling Service to help youth in crisis find employment and become self-reliant [24].

The present study determined the efficacy of a horticultural therapy program, based on the precepts of the TOP, that was designed to improve the emotional health and career exploration skills of adolescents at the D-JDC. The program utilized native plants to improve the self-efficacy, peer attachment, social skills, career preparation skills, and self-efficacy in career decision making among the juveniles at the JDC.

## 2. Materials and Methods

### 2.1. Research Subjects

We designed a horticultural therapy program with the help of an education officer at the D-JDC located in Daejeon city. We recruited male and female students who volunteered to participate. Before starting the study, the purpose of the study and precautions regarding horticultural therapy were explained, and informed consent was obtained from the volunteers. Thirty-five students completed the eight-session program, with an attendance rate of 91.4%. Owing to the nature of the JDC, which mainly provides psychological rehabilitation and treatment, the participants often missed sessions since they experienced frequent psychological changes, had to attend psychological counseling, or were released. The present study was approved by the Institutional Bioresearch Ethics Committee of Konkuk University (7001355-2018-HR-275).

### 2.2. Horticultural Therapy Program

The horticultural treatment program consisted of eight 90-min sessions of plant cultivation and harvesting activities once a week. *Spiraea salicifolia* L., *Hydrangea serrata* f. *acuminata* (Siebold & Zucc.) Wilson, *Caryopteris incana* (Thunb. ex Houtt.), *Allium senescens* L., *Iris ensata* ‘spontanea’ (Makino) Nakai, *Aster koraiensis* Nakai, *Lythrum salicaria* L., *Dendranthema indicum* (L.) Des Moul., *Pteridium aquilinum* (L.) Kuhn, *Mentha arvensis* ‘piperascens’ L., and *Prunella vulgaris var. lilacina* were used in various activities, such as seeding, transplanting plants, cutting, harvesting, decorating pressed flowers, and still life drawing (Table 1).

Self-awareness and career recognition, exploration, design, and preparation were broadly classified into four stages, and the goals for each session were planned according to the stages of career development and TOP guidelines. Session 1 focused on the self-awareness stage in which individuals aimed to understand native plants and express themselves. Session 2 focused on the planting stage in which individuals were encouraged to meet their mentor. Session 3 focused on the career awareness stage that involved planting and making a business card. Session 4 involved planting and caring for plants as well as understanding gardening-related occupations. Session 5 involved sowing seeds and searching for employment. Session 6 involved sowing seeds and experiencing work as an urban agriculture manager. Session 7 involved harvesting plants and experiencing work as a welfare gardener. Finally, session 8 focused on the design and preparation stage in which the participants prepared themselves to achieve their ambition. The program also included field activities (Table 2).

A horticultural therapist certified by the Korea Horticultural Therapy & Well-being Association, two assistants who were horticultural therapists, and two schoolteachers facilitated the program. In case of emergency, the teacher in charge of the JDC carefully observed the situation and called for medical interventions, as needed. The teacher in charge conducted a body search of the participants before and after each session to check for dangerous tools.

### 2.3. Measurements

The effectiveness of the horticultural therapy program was self-assessed before and after implementation. For emotional health evaluation, the ego-resiliency scale (ER), developed by Block and Kremen [25], was used; it consists of 14 items that are scored using a 5-point Likert scale, and the higher the summation score, the higher the individual’s ego resilience [25,26] (Cronbach’s α = 0.783).

For evaluating social behavior, the Inventory of Parent and Peer Attachment (IPPA), developed by Armsden and Greenberg [27], was used. It consists of 12 items that are scored by the adolescents using a 4-point Likert scale, and the higher the score, the higher the attachment to peers [28] (Cronbach’s α = 0.72). In addition, the SSS of the social skills rating system (SSRS), a US-standardized tool developed by Gresham and Elliot [29] was used to objectively evaluate the changes in social behavior. The SSS consists of 30 items that are scored on a 2-point Likert scale; it was completed by the teacher in charge [30] (Cronbach’s α = 0.85).

We used the career decision-making self-efficacy: short form (CDMSE-SF) developed by Lea and Lee [31]. The CDMSE-SF consists of 25 items that are scored using a 5-point Likert scale. Higher scores indicate that the person can successfully accomplish career decision-related tasks and make decisions carefully and wisely [32] (Cronbach’s α = 0.94). The career preparation behavior scale, developed by Park [33], was also used; it consists of 22 items that are scored using a 5-point Likert scale [34] (Cronbach’s α = 0.88). To assess the satisfaction of participants in the horticultural therapy program, a 10-question satisfaction questionnaire developed by the current research team was administered.

### 2.4. Data Analysis

For demographic data analysis, descriptive statistics were calculated (mean, standard deviation, and percentage of each item) using Microsoft Excel (Office 2010; Microsoft Corp., Redmond, WA, USA). To analyze the effects of the horticultural therapy program, the distribution normality was tested using SPSS (Version 25 for Windows; IBM, Armonk, NY, USA), and the data were analyzed using paired t-tests, with the significance level set at *p* < 0.05.

## 3. Results

### 3.1. Characteristics of the Subjects

The respondents were aged between 10 and 19 years (mean age, 15.74 [±1.65] years); 11 male and 24 female students were included. Regarding educational background, 97.1% had a middle school diploma or lower, 88.6% of them were students at the time of the survey, and 54.36% lived in the metropolitan area (Table 3).

### 3.2. Effects on Emotional Health

Table 4 shows the effect of the program on the ego resilience scores of the adolescents. The ego resilience score had increased from an average of 47.83 (10.09) before the program to an average of 49.37 (11.59) after the program (*p* = 0.365). Among the sub-items, vitality, interpersonal relationships, emotional control, curiosity, and optimism showed prominent changes.

### 3.3. Social Behavior

Table 5 shows the effect of the program on social skill scores. Peer attachment scores showed a significant improvement from the pre-score of 31.71 (4.99) to the post-score of 33.66 (5.52) (*p* = 0.05). Social skill scores also showed a significant improvement from the pre-score of 17.66 (15.30) to the post-score of 27.91 (15.03) (*p* < 0.001).

### 3.4. Career Preparation and Self-Efficacy

Table 6 shows the effect of the program on career preparation behavior and CDMSE-SF scores. Career preparation behavior score showed a significant improvement with an increase from the pre-score of 54.80 (17.14) to the post-score of 63.00 (23.39) (*p* < 0.05). The CDMSE-SF score significantly improved from the pre-score of 80.69 (21.04) to the post-score of 86.09 (20.57) (*p* = 0.113).

### 3.5. Satisfaction with the Program

Among the youths who participated in this study, 88% stated that they were generally satisfied with the program; furthermore, 94% stated that they were satisfied with the total number of sessions in the program (eight), and 6% stated that they were not satisfied. Regarding the number of sessions, the following responses were collected: 100 session (one youth), 12 (two youths), 10 (two youths), 3 (three youths), and 2 (one youths). Of the respondents, 74% (23 persons) stated that they were satisfied with the 90-min session duration; others preferred session durations of 10, 30, 60, 100, and 120 min. Regarding the frequency of once a week, 71% were satisfied and 10% were dissatisfied; 50% of those who were not satisfied wanted the program to run twice a week. The most preferred activity was cutting (16%) followed by flower decoration (15%), urban agriculture manager job experience (14%), planting (13%), welfare gardener job experience (13%), seed sowing (12%), harvesting (11%), and transplanting (6%). Further, 45% stated that it was good to learn new things and that the therapy enhanced their job search skills (76%). The respondents were open to participating in horticultural therapies in the future and were willing to recommend the therapy to their friends. They stated that the program was fun and friendly.

## 4. Discussion

The eight-session horticultural therapy program had a significant effect on the development of the peer attachment, social skills, and career preparation behaviors of the adolescents at the D-JDC. The adolescents reported a significant increase in peer attachment after the horticultural therapy program. This result was consistent with those of previous studies, where indoor horticultural therapy with easy access improved peer relationships through plant care and group activities and that horticultural activities had a positive effect on peer relationships [35,36]. Although the effects of horticultural therapy activities were subjective, objective increments in the vitality of the participants and the bonds between peers through cooperative group activities and plant care were observed.

Regarding social skills, the teachers noticed a synergistic effect of the therapy. The adolescents performed well in cooperative functions such as helping others, sharing objects, and following rules or instructions for using living plants. It is believed that positive effects were achieved by making self-assertions, accepting suggestions from friends, and exercising self-control in a conflict situation. These results were consistent with those of previous studies, where increased social relations were reported among children who engaged in horticultural therapy [37,38,39].

Career preparation behavior showed a significant improvement, consistent with the results of a previous study that reported on the ego-resiliency and social support of out-of-school adolescents [26]. Ginzberg [40] proposed that career development is a process that spans 6–10 years, beginning at about 11 years and ending at the age of 17 or in early adulthood [41]; hence, our horticultural therapy program focused on career exploration by offering opportunities for self-discovery and direct exploration. Preparing for employment is considered to be beneficial to the re-socialization process. In fact, job experience was the most preferred activity (27%) in the horticultural therapy satisfaction survey.

The students who completed the horticultural therapy program showed improved self-resilience and career decision self-efficacy and developed skills to cope well with internal and external stressors. The sub-items of ego-resiliency, interpersonality, emotional control, curiosity, optimism, and vitality showed significant improvements. The expectations and curiosity about new plants and activities observed in each session appear to support the problem-solving ability of the participants. This result is in line with those of a previous study where attachment to plants and curiosity in a cultivation-based horticultural therapy program were found to be effective in improving the psychological health of adult prisoners [42]; in another study, horticultural therapy was effective in improving ego-resiliency by stimulating children’s interest and curiosity [43,44,45].

The program had a positive effect on self-efficacy in career decision-making, which indicates that experiencing and understanding plants and plant cultivation firsthand enabled the participants to accomplish the horticultural-related career activities, which in turn increased their confidence in their ability to make careful and wise decisions. Considering that juvenile offenders have to leave the regular school system and receive correctional education at the JDCs, the educational function of the JDCs should be developed. Instead of solely pursuing responsibility for misconduct among the juvenile offenders, the state, as the new guardian, should also focus on character development, specialized education, and career development training to help them gain the knowledge and skills necessary for their social roles after release [6,46].

## 5. Conclusions

The present study verified that the emotional stability and social adaptability of the juveniles at the D-JDC changed positively through the implementation of a horticultural therapy that focused on career exploration for adolescents with uncertain career paths. Furthermore, the program was effective in improving the participants’ career design ability. This study is limited to certain region participants, and it is difficult to generalize the results due to the absence of a control group. Future studies should focus on further understanding the role of native plants in supplementary career-based education for juveniles. The horticultural therapy program developed in this study is expected to be an excellent educational program for improving career preparation among young people in correctional institutions.

## Figures and Tables

**Table 1 ijerph-19-08812-t001:** Horticultural therapy program using native plants for career exploration and emotional healing.

Session	Career Development Phase	TOP Phase	Horticultural Activity		Main Purpose	
					Career Education	Horticulture-Related Job Education
Orientation and pre-evaluation						
1	Self-assertion	Self-exploration	Understanding native plants	Self-expression		Life history of native plants Plant book preparation
2	Career assertion	Find ambition	Planting (*Lythrum salicaria*, *Dendranthema indicum, Allium senescens*)	Meet mentor		Basic knowledge of plant cultivation
3		Plan future	Planting (*Dendranthema indicum, Hedera helix*)	Design business card		Basic knowledge of plant cultivation
4	Career exploration	Express mind	Planting (*Aster koraiensis*, *Iris ensata* ‘spontanea’)	Understanding horticulture-related jobs		Basic knowledge of plant cultivation
5		Become rich	Seeding (*Caryopteris incana, Hydrangea serrata* f. *acuminata*)	Search for employment		Basic skills of plant cultivation
6		Self-marketing	Seeding (*Spiraea salicifolia*)	Job experience: urban agriculture manager		Basic skills of plant cultivation
7		Self-marketing	Pressing flowers after harvesting (*Pteridium aquilinum* ‘latiusculum’, *Mentha piperascens*)	Job experience: welfare gardener		Basic skills of plant cultivation
8	Career design and preparation	Future employment	Native plant utilization Decorate pressed flowers (*Pteridium aquilinum* ‘latiusculum’, *Mentha piperascens*)	Decision making and achieving ambitions		Basic skills of plant cultivation
Completion ceremony and post-evaluation						

**Table 2 ijerph-19-08812-t002:** Example of a session in the horticultural therapy program.

Title	Towards Dreams and Visions	Time Amount	13:00–16:40
Session	2	Place	D-JDC program room
Subject	Career education	Think about ambition	
	Horticulture education	Basic knowledge of plant cultivation; soil and transplantation	
Activity Goal	Understanding life–career development, finding ambition, and planning for the next steps. Build a foundation for growing native plants		
Materials	*Lythrum**salicaria, Dendranthema indicum*, Jifiport, bed soil (Pearlite, Pete Moss, Bark), seedling shovel, pot		
**Activity sequence**	**Activity**	**Activity intervention**	
Introduction (30 min)	1. Introduction and greeting 2. Contents and purpose of the session 3. Career awareness (ambition search) 4. Introduction to native plants and still life drawing book	1. Therapist and client, Clients and clients -Greetings and summarizing previous session 2,3. Learn about career development stages; learn about plant life cycle 4. Understand today’s plants and write a plant book	
Deployment (50 min)	1. Understanding current state and ambition awareness 2. Dream, vision, mentor 3. Soil and transplantation 4. Name tag	1,2. Understand current state, envision your ambitions, and set mentors for change. 3. Provide basic knowledge of soil -Observe sampled various types of soil and understand their function -Container soil: culture soil, mixing method -Outdoor Soil: Flower bed soil, adding organic matter -Transplantation and management 4. Name the plant	
Arrangement (10 min)	1. Care tips 2. Water cycle 3. Finish	1,2. Understand how to care for wildflowers 3. Arrangement and notice for the next session	
Notice			

**Table 3 ijerph-19-08812-t003:** Demographic information of participants (*n* = 35).

Variable	Category	Frequency (*n*)	Percentage (%)
Age	10–19 years old	35	100
Sex	Male	11	31.4
	Female	24	68.6
Final education	Elementary school	12	34.3
	Middle school	22	62.8
	High school	1	2.8
Current occupation	Student	31	88.6
	Office worker	0	0
	Agriculture and livestock	1	2.8
	Other	3	8.6
Marital status	Single	35	100
Monthly income	<1750 USD	2	5.7
	1750–3490 USD	4	11.4
	3490–5230 USD	4	11.4
	>5230 USD	0	0
	undeclared	25	71.4
Residence	Metropolitan area	19	54.3
	Chungcheong area	6	17.1
	Jeolla area	6	17.1
	Gyeongsang area	4	11.4

**Table 4 ijerph-19-08812-t004:** Effect of horticultural therapy program on ego-resilience scores.

Variable	N	Mean (SD)		*p* Value
		Pre-Test	Post-Test	
Human relationship	35	10.94 (2.13)	11.11 (2.78)	0.659
Vitality	35	6.83 (2.20)	7.37 (1.91)	0.105
Emotion control	35	6.29 (1.93)	6.46 (2.15)	0.676
Curiosity	35	17.09 (4.95)	17.43 (5.00)	0.628
Optimism	35	6.69 (1.82)	7.00 (1.75)	0.365
Total	35	47.83 (10.09)	49.37 (11.59)	0.365

SD, standard deviation; by paired *t*-test.

**Table 5 ijerph-19-08812-t005:** Social effects of horticultural therapy program.

Variable	N	Mean (SD)		*p* Value
		Pre-Test	Post-Test	
Peer attachment	35	31.71 (4.99)	33.66 (5.52)	0.016 *
Social skills	35	17.66 (15.30)	27.91 (15.03)	0.000 ***

SD, standard deviation; *, significant at *p* < 0.05; ***, significant at *p* < 0.001 by paired *t*-test.

**Table 6 ijerph-19-08812-t006:** Career preparation and self-efficacy effects of horticultural therapy program.

Variable	N	Mean (SD)		*p* Value
		Pre-Test	Post-Test	
Career preparation behavior	35	54.80 (17.14)	63.00 (23.39)	0.016 *
Career decision making self-efficacy	35	80.69 (21.04)	86.09 (20.57)	0.113

SD, standard deviation; *, significant at *p* < 0.05 by paired *t*-test.

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
