# Peer review of "Efficacy of a Horticultural Therapy Program Designed for Emotional Stability and Career Exploration among Adolescents in Juvenile Detention Centers"

_ijerph, 2022, doi:10.3390/ijerph19148812_

Round 1

Reviewer 1 Report

This manuscript outlines a horticultural intervention designed for and given to adolescents in a juvenile detention center (JDC), and has a number of strengths. The design of the horticultural program was excellent, and the evaluative tools used to assess impact of the study in a pre-post fashion were appropriate. 

There were a number of weaknesses that if addressed would improve the manuscript.

1) The participants were recruited from a single JDC. An acknowledgement of the potential lack of generalizability is warranted.

2) The participants were individuals who were interested in participating in the intervention. Since they were not a sampling of the adolescents from the JDC, the conclusions need to better reflect this self selection. A demographic summary of all adolescents in the JDC would also help judge the potential within-center representativeness (e.g. age and sex distributions from within the full population of the JDC).

3) An overall attendance percentage was reported. It would have been informative had a percentage been computed for each participant, and then summarized across the 35 participants.

4) Was the per-participant participation associated with changes in the outcome measures of interest?

5) I would have been interested in seeing summaries of the within-person change scores for the scales reported in tables 4 through 6 (e.g. by adding columns to the tables)

6) The conclusion that emotional stability was improved is over-stated, as there were not significant changes for the corresponding measures of "ego-resilience"

Reviewer 2 Report

The article, entitled “Development and Verification of a horticultural therapy pro-gram for emotional stability and career exploration of adolescents in juvenile detention centres”, studies the effects of horticultural therapy on juvenile delinquents.

The article is interesting, well written and consistent between introduction, analysis and discussion.

I suggest some improvements. For readers of other judicial systems in other countries, it would be good to include a mention of the juvenile offender treatment system in South Korea in the introduction, e.g.

- is there a specific court for juveniles or is it unique with adults?

- from what age a juvenile is imputable.

- among those who participated, it would be important to understand what type of offence they committed.

- whether there are differences between males and females and by age group, as there are major developmental changes between the ages of 10 and 19. The reseracher please, explain this choice.

Reviewer 3 Report

The work presented is of enormous interest and relevance. It is worth paying attention to the tools that can favor the insertion of adolescents who are in a situation of greater vulnerability, in this case adolescents who commit criminal offenses. 

However, there are some aspects that the authors could consider to improve the manuscript presented:

- At the end of the introduction, the objective of the work should be clearly incorporated. Right now it is confusing.

- The methodology is confusing. It would be advisable to explain on the one hand the program developed and on the other hand the aspects that are directly linked to the research study.

- The authors should include a section on limitations and future lines of work based on the study carried out.

- It does not seem that a program of only 8 sessions could have such relevant effects as those sought by the authors. 

Round 2

Reviewer 3 Report

The changes satisfy this reviewer's requirements